# Optical Fiber Sensor Performance Evaluation in Soft Polyimide Film with Different Thickness Ratios

**DOI:** 10.3390/s19040790

**Published:** 2019-02-15

**Authors:** Yanlin He, Xu Zhang, Lianqing Zhu, Guangkai Sun, Xiaoping Lou, Mingli Dong

**Affiliations:** 1Beijing Engineering Research Center of Optoelectronic Information and Instruments, Beijing Information Science and Technology University, Beijing 100192, China; heyanlin@bit.edu.cn (Y.H.); zhangxubistu@outlook.com (X.Z.); guangkai.sun@bauu.edu.cn (G.S.); louxiaopin@bistu.edu.cn (X.L.); 2Bionic and Intelligent Equipment Lab, Beijing Information Science and Technology University, Beijing 100192, China

**Keywords:** soft robotics sensor, sensitivity, micro curvature sensor, polyimide film, embedded depth

## Abstract

To meet the application requirements of curvature measurement for soft biomedical robotics and flexible morphing wings of aircraft, the optical fiber Bragg grating (FBG) shape sensor for soft robots and flexible morphing wing was implemented. This optical FBG is embedded in polyimide film and then fixed in the body of a soft robot and morphing wing. However, a lack of analysis on the embedded depth of FBG sensors in polyimide film and its sensitivity greatly limits their application potential. Herein, the relationship between the embedded depth of the FBG sensor in polyimide film and its sensitivity and stability are investigated. The sensing principle and structural design of the FBG sensor embedded in polyimide film are introduced; the bending curvatures of the FBG sensor and its wavelength shift in polyimide film are studied; and the relationship between the sensitivity, stability, and embedded depth of these sensors are verified experimentally. The results showed that wavelength shift and curvature have a linear relationship. With the sensor’s curvature ranging from 0 m^−1^ to 30 m^−1^, their maximum sensitivity is 50.65 pm/m^−1^, and their minimum sensitivity is 1.96 pm/m^−1^. The designed FBG sensor embedded in polyimide films shows good consistency in repeated experiments for soft actuator and morphing wing measurement; the FBG sensing method therefore has potential for real applications in shape monitoring in the fields of soft robotics and the flexible morphing wings of aircraft.

## 1. Introduction

In an effort to implement the sensing and monitoring of soft biomedical robotics and the flexible morphing wings of aircraft, researchers have studied their applicable sensors and characteristics [1,2,3]. Some general rigid sensors, such as strain gauges and encoders [4,5], are incompatible with soft robots and flexible morphing wings [6,7,8] due to the complex application environment and various obstacles, such as the complex cable networks and electromagnetic interference. Visual sensing methods are not suitable for soft medical robots in narrow spaces and morphing wing sensing during flight. As a result, in recent years some soft sensors and sensing methods that can be used for structural sensing and monitoring have been desired [9,10,11].

There are unique advantages to optical fiber sensors; they are light-weight, small size, having low transmission loss, instant response, immunity to electromagnetic interference, and being able to be conveniently placed in the interior of a given substance [12,13,14,15,16]. In addition, unlike the traditional fabrication process of ordinary electrical sensors, the manufacturing technology of optical fiber is simple, and produced with a single fiber. Furthermore, optical fiber Bragg grating (FBG) sensors are well developed and widely used. More importantly, optical fiber FBG sensors are harmless to patients when used as an in vivo sensor. Therefore, when these sensors are embedded into the patient’s body, they can implement inherently safe sensing of the patient, and several studies on FBG-based sensors have been conducted [17,18,19,20,21].

T. C. Searle et al. [22] proposed a measurement system for the soft manipulator that used optical fiber for constructing its sensing network. The sensing mechanism of optical fiber is based on laser power modulation, and the sensor developed in the aforementioned study has good performance for the bending angle and orientation measuring, as well as the bending radius measurement of a soft manipulator. Chen et al. [23] designed a polymeric grating sensor, and the core of the grating sensor deviates from the central axis of the fiber. The experimental results of the aforementioned study demonstrated that the polymeric grating sensor was able to test the strain, bending, temperature, and asymmetric structure that could enhance the sensitivity of the sensor. A. S. Silva et al. [24] proposed a sensing system for a biomedical garment that embedded fiber sensors into the garment. Via angle evaluation of the joint, the motion performance of the patient was tested and monitored. For the patient’s elbow motion, this sensing structure is comfortable and has high test performance. S. Sareh et al. [25] proposed a pose measuring system for soft robotic arms, which embedded micro optical sensors in its body, and—in the aforementioned study—optical fibers were coupled in a basal unit and integrated with a distance modulation array. Through the structural change and voltage measurement results, the posture of the soft arm can be obtained. Ge et al. [26] proposed an FBG-based curvature sensor in the soft silicone material, and the FBG sensor was embedded off-center to the soft silicone. Through the obtained wavelength shift of the FBG sensor, this soft silicone sensor is capable of distinguishing the bending directions and curvatures of the object, and its measurement range extends to ±80 m^−1^. H. Zhao et al. [27] designed a sensing system for soft orthosis based on embedded optical fibers. A part of the cladding layers was removed from one side of fibers, the power of the laser transmitted through the optical fiber decreased, and the laser dissipation was greatly interrelated to the soft orthosis’s curvature. L.J. Arnaldo et al. [28] proposed a Polymer Optical Fiber Bragg Gratings (POFBG) curvature sensor inscribed in cyclic transparent amorphous fluoropolymers fibers. The development yielded a reduction of both mean squared errors and hysteresis, and the frequency cross-sensitivity and hysteresis of the sensor was implemented through a compensation algorithm. In our previous study [29], an optical fiber FBG shape reconstruction of a soft actuator was realized. For the sake of increasing compatibility between the FBG sensor and soft actuator, the FBG sensor was glued to polyimide film and encapsulated by the silicone layer. However, the sensitivity and stability of the FBG sensor and the relationship of its embedded depth in polyimide film were not mentioned.

Unlike these previous studies, the sensitivity and stability of the flexible optical fiber sensor,—which is embedded in polyimide film at different depths,—is studied herein. Polyimide film could increase the compatibility of the optical fiber sensor and soft robotics. The relationship between the bending curvature of the FBG sensor in polyimide film and its wavelength shift was theoretically analyzed, and the sensitivity and stability of the FBG sensor with various embedded depths in polyimide film was verified experimentally.

For the remaining sections of this paper in Section 2, the design and sensing principle of the polyimide FBG sensor is described. Section 3 presents sensing experiments using the sensor in polyimide film at different embedded depths. The interrelation to the curvature of the FBG sensor and its wavelength shift, as well as its sensitivity, is discussed in this section. Finally, Section 4 provides our conclusions.

## 2. Materials and Methods

### 2.1. FBG Sensor Embedded in Polyimide Film

Figure 1a presents the proposed FBG sensor embedded in polyimide film and its measuring setup. Through bonding two polyimide tapes together, the optical FBG sensor is embedded in polyimide film and is well-protected. The FBG sensor is located on the y-axis of the polyimide film, and deviates from the z-axis. Figure 2 presents the physical layout of the FBG sensor in the polyimide film and its calibration blocks. As we can see, the polyimide tapes are close together with the FBG sensor in its bending state. Owing to the flexible characteristic of polyimide tape and optical fiber, the FBG sensor embedded in polyimide films can be placed onto test objects and complies with the curvature profiles of the test objects perfectly.

In this paper, the FBG sensor’s diameter is 0.25 mm, and the height of a single polyimide film is 0.1 mm. We integrated ten polyimide film layers together to form the curvature sensor. The height (h_1_ + h_2_) of the polyimide film sensor is approximately 1.25 mm, the width of the sensor is 10 mm, and the length of the sensor is 30 mm. The polyimide FBG sensor applied in the optical sensor is a normal FBG manufactured with a UV writing method. For the sake of exploring the sensitivity and stability of FBG sensors in polyimide film with different embedded depths, we fabricated five sets of sensors. The thicknesses of the upper and lower polyimide film layer of these five sensors are in the ratio 0.9:0.1, 0.8:0.2, 0.7:0.3, 0.6:0.4, and 0.5:0.5, for Nos. 1, 2, 3, 4, and 5, respectively. Considering that the pre-tightening force and the annealing time of the FBG are slightly different during the writing process, the center wavelength of optical FBG has slight differences. Therefore, the spectral peaks of these five sets of FBG sensors are approximately 1547.6037 nm, 1547.9260 nm, 1547.8623 nm, 1547.5900 nm, and 1547.5910 nm, for Nos. 1, 2, 3, 4, and 5, respectively. The grating length, reflectivity, and side-mode suppression of all five sensors are 10 mm, 90%, and 20 dB [18].

### 2.2. Sensing Theory of the Polyimide FBG Sensor

In line with the sensing theory of the FBG sensor, the wavelength of the FBG sensor is obtained from the period of grating, which can be written as follows [27,30,31]:(1)$$\lambda_{B}=2n_{eff}\times \Lambda$$
where *n_eff_* represents the refractive index of fiber and *Λ* represents the period of grating.

Due to the changes in the strain of the grating and the temperature, the wavelength of the FBG sensor and its grating period also varied. The interrelation between the grating strain, wavelength shift, and temperature variation are as follows:(2)$$\Delta \lambda_{B}/\lambda_{B}=\left(1-P_{\varepsilon }\right)\cdot \varepsilon +\left(\alpha_{T}+\xi \right)\cdot \Delta T$$
where Δ*λ_B_* represents the wavelength variation, *ε* represents the soft actuator’s strain, *P_ε_* is a constant that is determined by the optical fiber and the coefficient of photo-elastic, *α_T_* is the thermal coefficient expansion of grating, *ξ* represents the thermo-optical coefficient of grating, and Δ*T* represents the temperature variation.

Assuming that the FBG sensor in polyimide film functions at a constant room temperature, the influence of working temperature can be ignored in this study. Thus, Formula (2) can be converted to Formula (3). From the wavelength shift and coefficient of photo elastic, the strain of the FBG sensor can be obtained with the following formula:(3)$$\Delta \lambda_{B}/\lambda_{B}=\left(1-P_{\varepsilon }\right)\cdot \varepsilon$$

In this study, given that the FBG sensor is located in polyimide film, the FBG sensor and polyimide film can be considered as the same structure. Under an actuating force, the FBG sensor and polyimide film bear a uniform deformation. Figure 2 illustrates the embedded depth and bending deformation of the FBG sensor in polyimide films; the length and thickness of the FBG sensor in polyimide film is denoted by *L* (30 mm) and *h* (1.25 mm).

According to a previously determined model [25,28,29], the FBG sensor is located deviation from the central axis, as presented in the dashed line E–F in Figure 2b. *h*_1_ represents the range from the FBG sensor to the outer arc of polyimide film, and *h*_2_ represents the range from the FBG sensor to the inner arc of the polyimide film. When polyimide film and its embedded FBG sensor are deformed as Figure 2b presents, the interrelation to the bending radius, bending angle, and the length of the central axes are determined by
(4)$$L_{EF}=R\cdot \theta$$
where *L_EF_* represents the polyimide film’s central axis length, *R* is the bending radius, and *θ* is the bending angle of the polyimide FBG sensor.

In the state of bending, we assume that length changes of the inner and outer part of the polyimide film are the same, and the FBG sensor is compressed under bending conditions. A slight modification in the bending radius of the FBG sensor caused by the offset location (*x*) is represented by; *R* − *x*. Under a bending curvature, the FBG sensor’s length (*L_AB_*) is expressed as follows:(5)$$L-\Delta L=L_{AB}=\left(R-x\right)\cdot \theta =\left(R-\left(h_{1}+h_{2}/2\right)\right)\cdot \theta$$
(6)$$L+\Delta L=L_{GH}=\left(R+x\right)\cdot \theta =\left(R+\left(h_{1}+h_{2}\right)/2\right)\cdot \theta$$
(7)$$x=h_{1}-\left(h_{1}+h_{2}\right)/2$$
where *x* is the range from the central axis to the FBG sensor. The change in length of the polyimide film induced by bending of the FBG sensor is derived as Equations (5) and (6), and the FBG sensor’s bending curvature in polyimide film can be expressed as follows:(8)$$C=\frac{1}{R}=\frac{2}{h}\frac{\Delta L}{L}=\frac{2}{h}\varepsilon =\frac{\Delta \lambda_{B}}{\lambda_{B}\left(1-R_{\varepsilon }\right)\cdot h}$$
where *C* represents the FBG sensor’s bending curvature in polyimide film, *R* represents the radius, Δ*L* represents the length change of the inner and outer part of the polyimide film, and *ε* represents the polyimide film’s strain. With regard to FBG sensors, $\lambda_{B}\begin{array}{cc} , & R_{\varepsilon }\begin{array}{cc} , & h \end{array} \end{array}$ are constant values and determined by the writing process. Furthermore, a linear relationship exists between wavelength shift and bending curvature.

## 3. Results and Discussion

### 3.1. Sensitivity and Stability of the Sensor

To evaluate the sensitivity of the FBG sensor in polyimide film at different embedded depths, five sensors were manufactured (No. 1 to No. 5) with various depths; the thicknesses of the upper and lower polyimide film of these five sensors were 9:1 (No. 1), 8:2 (No. 2), 7:3 (No. 3), 6:4 (No. 4), and 5:5 (No. 5). Figure 3 presents the experimental setup of these sensors, which consists of a broadband laser source, a spectrograph, a demodulator, and seven standard curvature calibration blocks. The central wavelength of the five sensors was approximately 1547.6037 nm, 1547.9260 nm, 1547.8623 nm, 1547.5900 nm, and 1547.5910 nm, respectively. A broadband light source (Lightpromotech M1043-13, Lightpromotech, Beijing, China) with wavelengths from 1529 nm to 1605 nm was launched into the FBG as the input light. A Yokogawa AQ6370C spectrograph with a wavelength measurement range of 600–1700 nm and a power range of −90 dBm to +20 dBm reflected the FBG signals. The demodulator used in this study was developed by our team, its wavelength ranged from 1525 nm to 1610 nm, and the rate of demodulation was 35 kHz.

In the experiment, seven curvature calibration blocks with curvatures of 0 m^−1^, 5 m^−1^,10 m^−1^, 15 m^−1^, 20 m^−1^, 25 m^−1^, and 30 m^−1^ were calibrated and tested. The FBG sensor in the polyimide film was bent and fit tightly onto the boundary of standard calibration blocks; the blocks and polyimide film were considered to have equal curvature. Some results for the shift in the wavelength of the polyimide film curvature sensor are given in Figure 4; all experiments were repeated five times.

Figure 4 shows that when the curvature of FBG sensor in the polyimide film ranges from 0 m^−1^ to 30 m^−1^, the wavelengths of these five sensors vary and their reflection peaks also shift gradually to a longer wavelength direction. For sensor No. 1, the center wavelength shifted from 1547.6037 nm to 11549.1488 nm; for sensor No. 2, it shifted from 1547.9260 nm to 1548.7170 nm; for sensor No. 3, from 1547.8623 nm to 1548.3081 nm; for sensor No. 4, from 1547.5900 nm to 1547.8318 nm; and, finally, for sensor No. 5, from 1547.5910 nm to 1547.6516 nm. Thus, the sensors’ relative wavelength changes are 1.5451 nm, 0.7910 nm, 0.4458 nm, 0.2418 nm, and 0.0606 nm. When the curvature of these sensors was the same, their wavelength shifts were determined by the depth at which they were embedded in the polyimide films, which successively governed the sensor’s sensitivity. Figure 5 illustrates the change relation of curvatures and wavelength shifts of FBG sensors in polyimide film; when the sensor’s curvatures range from 0 m^−1^ to 30 m^−1^ their wavelengths shift, and curvatures have an approximately linear relationship.

Figure 6 presents the sensitivities of the five FBG sensors with different embedded depths and shows that the sensitivity of curvature sensors in polyimide film increases as h_1_−h_2_ values increase. The maximum sensitivity of the polyimide film curvature sensor was 50.65 pm/m^−1^ and was obtained from the embedded depth of h_1_:h_2_ = 9:1. The minimum sensitivity of the polyimide film curvature sensors was 1.96 pm/m^−1^ and was obtained from the embedded depth of h_1_:h_2_ = 0.5:0.5.

We then carried out five groups of experiments to evaluate the stability of the sensor based on the polyimide film. In this study, the deviation index (DI) is defined to represent the stability of sensors and is expressed as follows:(9)$$D_{I}=\Delta \lambda_{E}/\Delta \lambda_{w}\times 100\%$$
where $\Delta \lambda_{E}$ represents the difference between each experimental result and the average wavelength of FBG sensors and $\Delta \lambda_{W}$ represents the entire wavelength shift.

The experimental results for stability of the curved polyimide film sensors are given in Figure 7. The fluctuating range of the sensor with an embedded depth of h_1_:h_2_ = 9:1 is −1.97–2.53%, and its fluctuation interval is 4.5%. The range of the fluctuation of the polyimide film curvature sensor with an embedded depth of h_1_:h_2_ = 5:5 is −20–32.71%, and its fluctuation interval is 52.71%. This indicates that the stability of the latter is better than the former and has a smaller fluctuation, and that a polyimide film curvature sensor with larger h_1_:h_2_ values has good stability.

### 3.2. Evaluation of the Polyimide FBG Sensor in a Soft Actuator

For the sake of performance evaluation of the polyimide FBG sensor in a practical application, the FBG sensor was embedded in a soft actuator, and some experiments were conducted when the curvature of the actuator increased from 0 m^−1^ to 30 m^−1^. The structure of the actuator is given in Figure 8; the FBG sensor is embedded into it with polyimide tape, and the thicknesses of the upper and lower polyimide film layers are 1:9. The soft actuator is controlled by input pressure to bend at different angles, and the pressure rate is 0.2 Hz.

The experimental results of the polyimide FBG sensor are shown in Figure 9, as the curvature of actuator varies with 0 m^−1^, 5 m^−1^, 10 m^−1^, 15 m^−1^, 20 m^−1^, 25 m^−1^, and 30 m^−1^. Its power peaks increased from 1544.498 nm to 1545.956 nm. Figure 9 also demonstrates that the peak wavelength slowly drifts to the larger wavelength direction, and the polyimide FBG sensor’s wavelength shift was approximately 1.458 nm. The curvature change of the polyimide FBG sensor is instantly sensitive to the bending state of the soft actuator.

Figure 10 presents the relationship between the curvature of the FBG sensor and its wavelength shift in the soft actuator. It indicates that the FBG sensor’s wavelength shift enlarged as the soft actuator’s curvature increased. Furthermore, with the curvature of 30 m^−1^, the maximum sensitivity of the polyimide FBG sensor in the soft actuator is 50.15 pm/m^−1^, and the fluctuating interval of the polyimide FBG sensor is 4.67%; this may be due to the flexible soft silicone and may have influenced the performance of the sensor. The polyimide FBG sensor designed in this study performs well in terms of measuring the curvature of the soft actuator and has a wide application potential to be used in curvature sensing in soft robotics.

### 3.3. Polyimide FBG Sensor Evaluation for a Morphing Wing

To evaluate the performance of the proposed polyimide FBG sensor in a new practical application, it was embedded in a polyimide film that was glued on the surface of a morphing wing of aircraft. Some experiments were performed in which the wing’s curvature was increased from 0 m^−1^ to 20 m^−1^. The structure of the morphing wing and its polyimide FBG sensor is shown in Figure 11, which shows the polyimide FBG sensor is located along the rib of the morphing wing.

As the wing’s curvature varied with 0 m^−1^, 5 m^−1^, 10 m^−1^, 15 m^−1^, and 20 m^−1^, the experimental results of the polyimide FBG sensor are shown in Figure 12; its power peaks increased from 1547.497 nm to 1548.529 nm, its wavelength shift was about 1.032 nm, and the curvature change of the polyimide FBG sensor was instantly sensitive to the morphing state of wing. Figure 13 presents the relationship between the curvature of the FBG sensor and its wavelength shift; the sensor’s wavelength shift enlarged as the wing’s curvature increased. With the wing’s curvature ranging from 0m^−1^ to 20 m^−1^, the maximum sensitivity of the polyimide FBG sensor is 64.14 pm/m^−1^, and the fluctuating interval of the polyimide FBG sensor is 4.17%. Consequently, the polyimide FBG sensor designed in this study performs well in terms of measuring the curvature of the morphing wing of aircraft.

## 4. Conclusions

To summarize, to meet the requirements for a tool used to measure the curvature of a soft biomedical manipulator and flexible morphing wings of aircraft, the objective of this study was to analyze a soft curvature sensor based on an embedded FBG sensor in a flexible polyimide film. The sensitivity and stability of this soft curvature sensor with an FBG sensor embedded at different depths in a polyimide film were investigated.

The FBG was embedded by binding two thin polyimide films together, so that the optical fiber and the FBG were well protected. The relationship between the embedded depth of the FBG sensor in the polyimide film and its sensitivity and stability were investigated; the wavelength shift and bending curvatures of the FBG sensor embedded in polyimide film were studied; and the relationship between the sensitivity, stability, and embedded depth of these sensors were verified experimentally. Experimental results illustrated that the maximum and minimum sensitivity of the polyimide FBG sensors were 50.65 pm/m^−1^ and 1.96 pm/m^−1^, with the curvature ranged up to 30 m^−1^. The designed FBG sensor embedded in polyimide films showed good consistency in repeated experiments for soft actuator and morphing wing measurement; the FBG sensing method therefore has potential for real applications in shape monitoring in the fields of soft robotics and flexible morphing wing of aircraft. In the future, we will focus on the design of multiple FBGs that can sense the shapes of non-rigid objects with morphological details in complex and unstructured environments, particularly for real-time sensing for soft surgical manipulators with biomedical applications and flexible morphing wings of aircrafts.

## Figures and Tables

**Figure 1 sensors-19-00790-f001:**
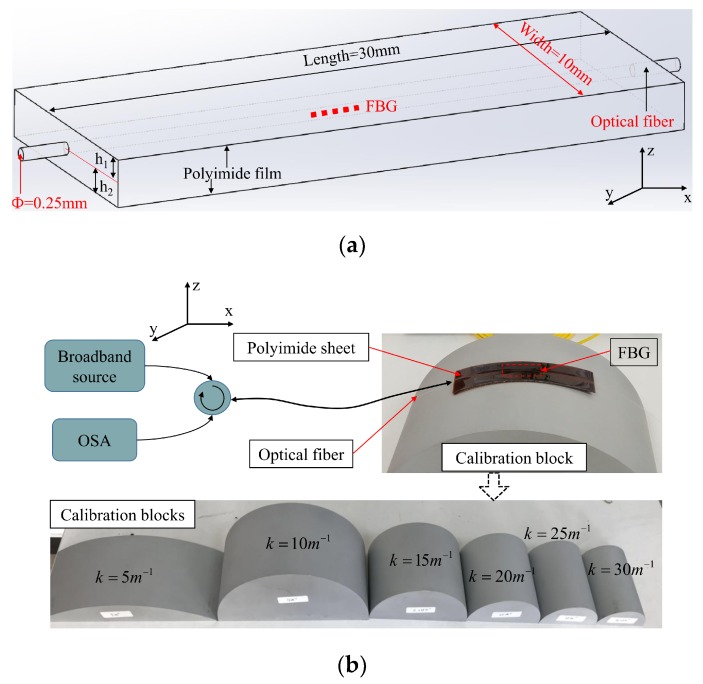
Physical layout of the optical fiber Bragg grating (FBG) sensor embedded in polyimide films. (**a**) Layout of the FBG sensor embedded in two polyimide films; (**b**) FBG sensor and its calibration blocks.

**Figure 2 sensors-19-00790-f002:**
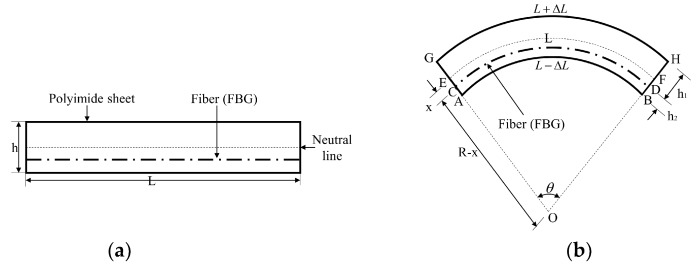
Sensing theory of the polyimide FBG sensor. (**a**) Free-state without bending and (**b**) bending state.

**Figure 3 sensors-19-00790-f003:**
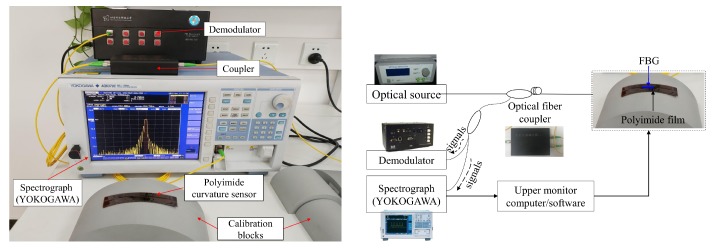
Setup used to study the polyimide film curvature sensor.

**Figure 4 sensors-19-00790-f004:**
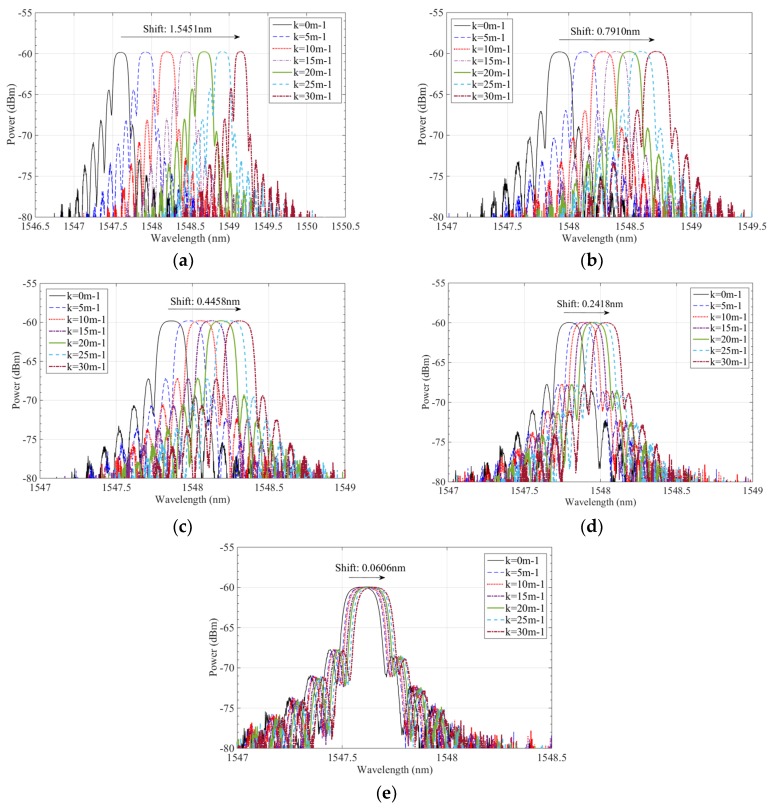
Wavelength shifts and intensities of FBG sensors with different thicknesses. (**a**) Test of sensor No. 1; (**b**) test of sensor No. 2; (**c**) test of sensor of No. 3; (**d**) test of sensor of No. 4; and (**e**) test of sensor of No. 5.

**Figure 5 sensors-19-00790-f005:**
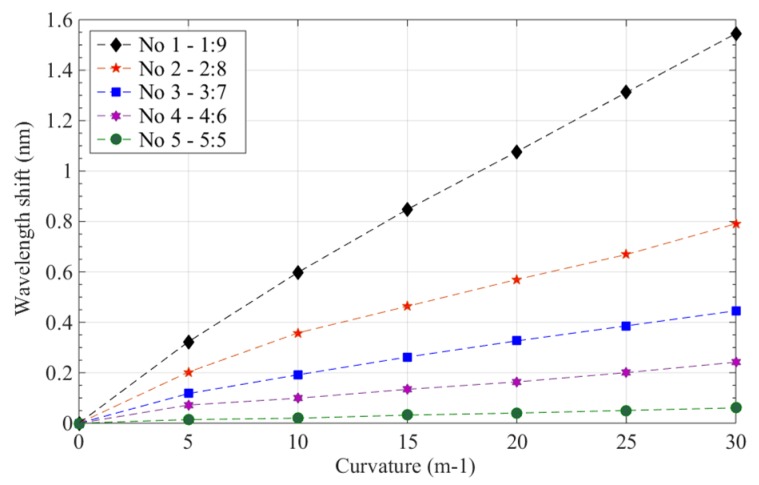
Wavelength shifts of FBG sensors with various curvatures.

**Figure 6 sensors-19-00790-f006:**
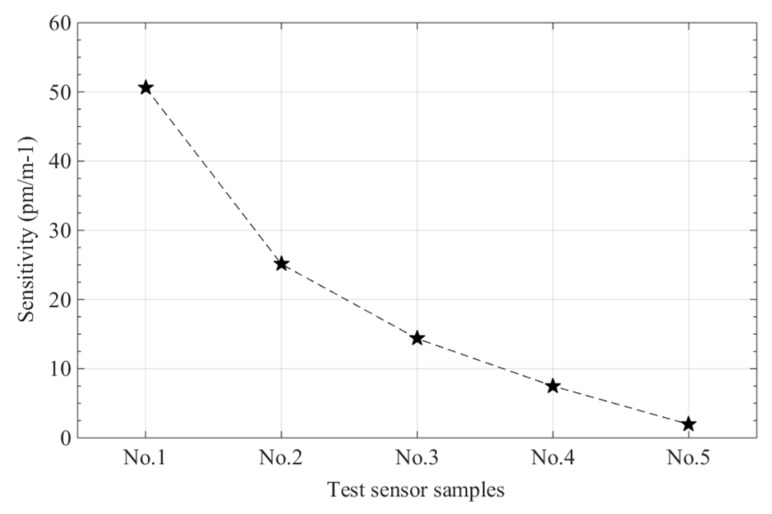
Sensitivity of fiber Bragg grating sensors at different embedded depths.

**Figure 7 sensors-19-00790-f007:**
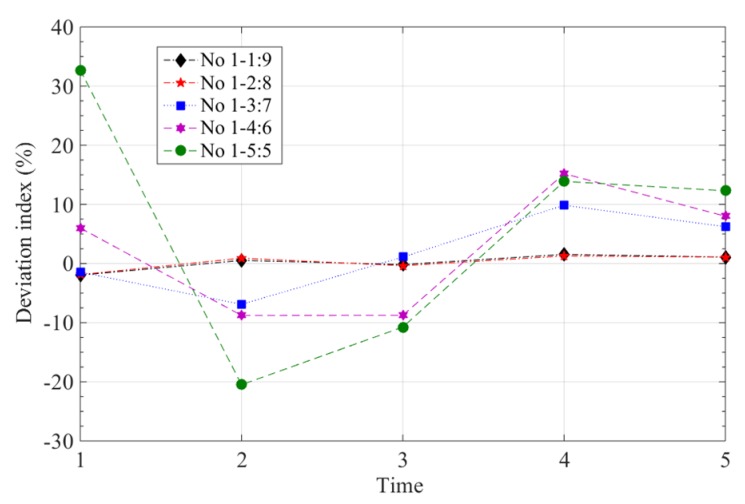
Results of tests of stability of polyimide film curvature sensors with different embedded depths.

**Figure 8 sensors-19-00790-f008:**
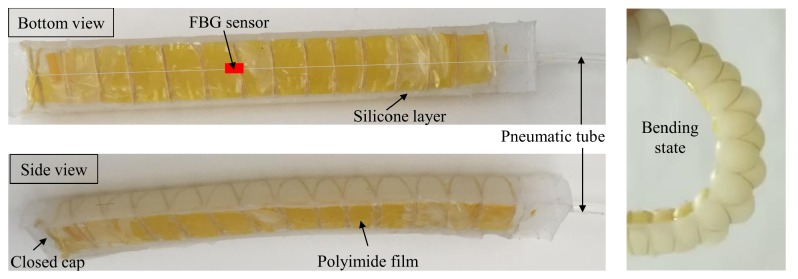
Polyimide FBG sensor application in a soft actuator.

**Figure 9 sensors-19-00790-f009:**
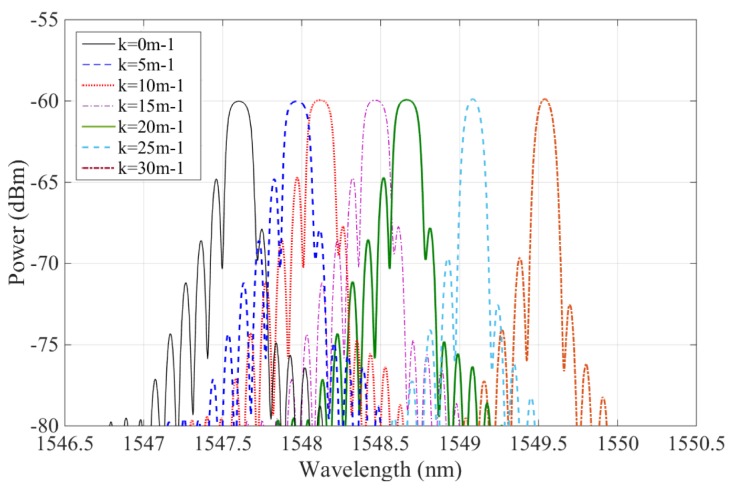
Wavelength shift and intensity of FBG sensor in a soft actuator.

**Figure 10 sensors-19-00790-f010:**
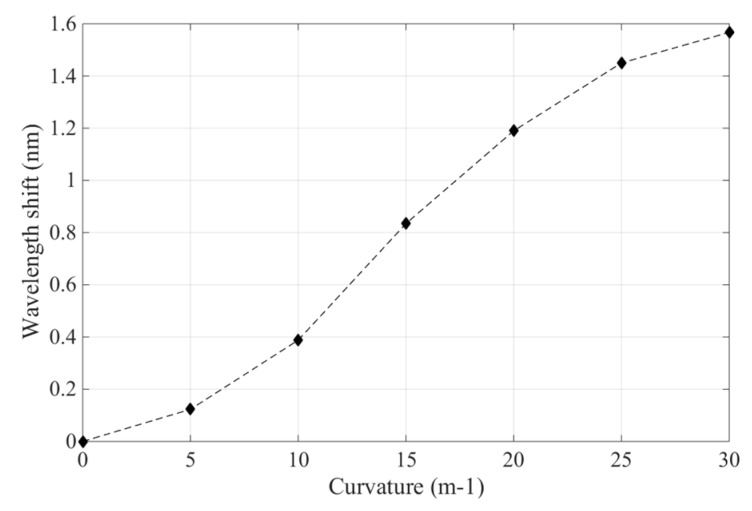
The wavelength shift of FBG sensor in a soft actuator.

**Figure 11 sensors-19-00790-f011:**
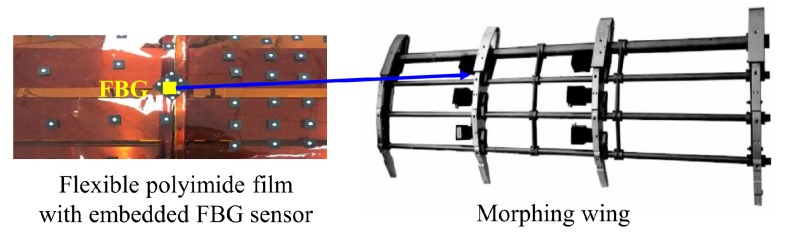
Polyimide FBG sensor in the morphing wing of an aircraft.

**Figure 12 sensors-19-00790-f012:**
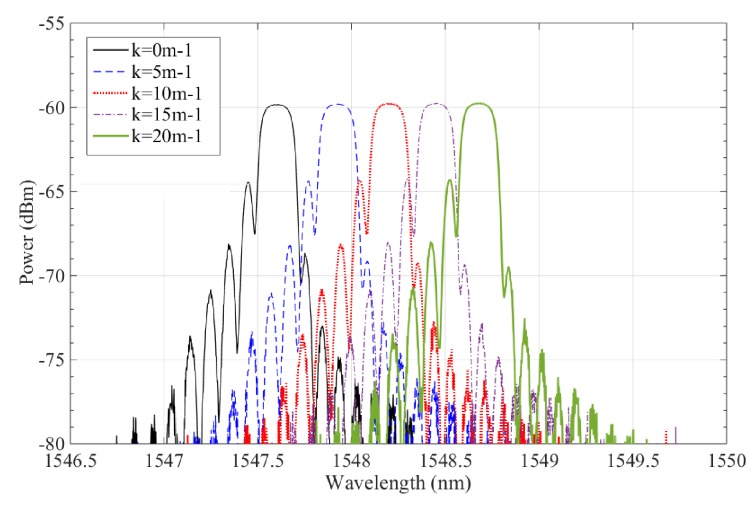
Wavelength shift and intensity of FBG sensor for a morphing wing.

**Figure 13 sensors-19-00790-f013:**
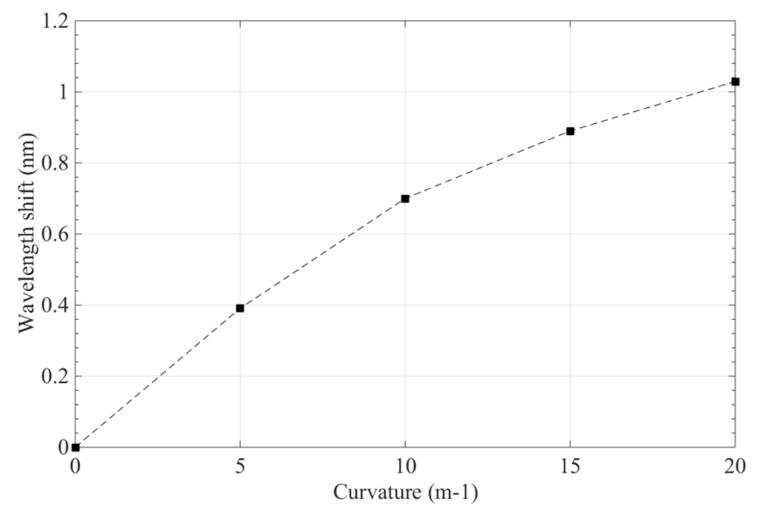
The wavelength shift of FBG sensor for a morphing wing.

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
