# Peer review of "Optical Fiber Sensor Performance Evaluation in Soft Polyimide Film with Different Thickness Ratios"

_sensors, 2019, doi:10.3390/s19040790_

Round 1
Reviewer 1 Report
The paper presents Fiber Brag Grafting (FBG) shape sensing for specific applications in soft robots and actuators applications. The manuscript has not shown any significant improvement or novelty compared to other similar work in the development of FBG sensors. It would be more acceptable if the authors could include a new application or a new fabrication method in this study.
Author Response
Dear Reviewer,
We would express our sincere appreciations for your insightful comments to the manuscript entitled “Preliminary Study on Soft Optical Fiber Sensor based on Polyimide Film” (Manuscript ID: sensors-423303). With your constructive suggestions in mind, revision has been completed to address your concern. We are confident that the quality of this paper has been improved after this revision, and some revised details are attached with files.

Reviewer 2 Report
Authors presented a FBG embedded in polyimide films for curvature sensing. Even though FBG embedment in polymer films was already explored in many works reported in the literature, the paper has merits regarding the study of the film thickness ratio. However, the paper needs a major revision in which the authors should comply with the comments listed below:
- First of all, a "preliminary study" as mentioned on the title does not provide sufficient information about the work reported on the paper. I recommend the authors change the title to include more information about the work, which can be done by removing the "preliminary study" and including something related to the evaluation of the film thickness ratio, for example.
- Authors should include in the introduction works made with FBGs inscribed in POFs for curvature sensing [A, B], since the POFs are more flexible and curvature sensing can be performed without the addition of protective films.
[A] Leal-Junior, A.; Theodosiou, A.; Díaz, C.; Marques, C.; Pontes, M.; Kalli, K.; Frizera-Neto, A. Polymer Optical Fiber Bragg Gratings in CYTOP Fibers for Angle Measurement with Dynamic Compensation. Polymers (Basel). 2018, 10, 674, doi:10.3390/polym10060674.
[B] Chen, X.; Zhang, C.; Webb, D. J.; Peng, G. D.; Kalli, K. Bragg grating in a polymer optical fibre for strain, bend and temperature sensing. Meas. Sci. Technol. 2010, 21, doi:10.1088/0957-0233/21/9/094005.
-Authors should provide a broader discussion about the methods employed on the FBG embedment on the polyimide film, e.g., how the film dimensions were controlled? What was the method used on the embedment? What is the uncertainty regarding the film dimensions?
- Authors should consider merging Tables 1-5.
- Authors should provide much more tests and evaluations on the sensors. The authors should include a comparison of the sensors linearity and hysteresis.
- In the whole introduction, the main motivation was the development of a sensor capable of meeting the requirements of soft robotics. However, in general, robots are dynamic devices working on a range of velocities. Thus, the sensor evaluation must be made with dynamic tests with predefined velocities, instead of just positioning the sensors on different calibration blocks.
- If possible, it would be nice if the authors employ the proposed sensor in a soft actuator in order to evaluate the sensor in a practical application.
- Polyimide is a polymer, which has viscoelastic nature. For this reason, the effect of temperature and movement frequency have influence on the material properties. Since the FBG is embedded in a polymer film, tests in different temperature and frequencies can influence the sensor sensitivity as shown in [A] (which also highlight the necessity of the sensor's assessment in dynamic tests). The authors should discuss this issue.
Author Response
Dear Reviewer,
We would express our sincere appreciations for your insightful comments to the manuscript entitled “Preliminary Study on Soft Optical Fiber Sensor based on Polyimide Film” (Manuscript ID: sensors-423303). With your constructive suggestions in mind, revision has been completed to address your concern. We are confident that the quality of this paper has been improved after this revision,,and some revised version are attaches with files.

Reviewer 3 Report
The paper is poorly English written and needs full revision. Too many commas, long sentences, and repetition of the same words (i.e. optical FBG sensor). Scientific content must be improved. Major revisions are required.
Comments:
Experimental setup presented in figure 1 is not clear. Two fibre couplers are used and should be stated in the text.
“Optical FBG sensor based on polyimide film” is misleading – the FBG is embedded in a polyamide film.
Tables 1 to 5 are unnecessary – figure 6 is enough.
On page 2, last paragraph, “the dimension of the optical FBG sensor is 0.25mm” but at the end of said paragraph, “The grating length, (…) of these sensors are 20 mm, …” – revise this.
What is the purpose of using such a long FBG?
What was the reason for choosing polyamide (instead of other material) for embedding the FBG?
Results from figure 6: non-linear response is observed, therefore, the sensitivities provided, assuming linear fit, must be revised. Also, sensitivities of all sensors (for a linear range) must be referred in the text.
The authors should properly explain what is “the stability and consistency” of the sensor. Also, the meaning of “smaller and larger fluctuations”. Results presented in figure 8 are not properly explained. Authors must provide significance of these results in the text.
Author Response

(The authors gave the same response as above.)

Round 2
Reviewer 1 Report
I yet have to see the application of the proposed sensor in the same paper.
Author Response
Dear Reviewer,
We would express our sincere appreciations for your insightful comments to the manuscript entitled “Preliminary Study on Soft Optical Fiber Sensor based on Polyimide Film” (Manuscript ID: sensors-423303). With your constructive suggestions in mind, revision has been completed to address your concerns. We are confident that the quality of this paper has been improved after this revision. Some details of the comments are attached with file.

Reviewer 2 Report
Authors addressed almost all my comments, due to the novel application in soft actuator. In my opinion, this paper is suitable for publication in MDPI Sensors.
Author Response
Dear Reviewer,
Thanks for your comments again!
Reviewer 3 Report
The manuscript has been significantly improved and it warrants publication in Sensors.
Author Response

(The authors gave the same response as above.)
